# High-Resolution Greening Scenarios for Urban Climate Regulation Based on Physical and Socio-Economical Factors

**Daniele La Rosa** [1,*] and **Junxiang Li** [2]

1   Department of Civil Engineering and Architecture, University of Catania, Via S. Sofia 64, 95125 Catania, Italy
2   Department of Landscape Architecture, School of Design, Shanghai Jiao Tong University,
    Shanghai 200240, China; junxiangli@sjtu.edu.cn
*   Correspondence: dlarosa@darc.unict.it

**Abstract:** Urban ecosystems represent the main providers of ecosystem services in cities and play a relevant role, among the many services, in the regulation of the urban microclimate and mitigation of the urban heat island effect. The amount, localization, and spatial configuration of vegetation (i.e., urban trees) are key elements for planners and designers aiming at maximizing the climate regulation potential and therefore extending the related benefits to a higher number of residents and city users. Different factors and constraints related to urban morphology and socio-economical characteristics of the urban environment influence the localization of new greening scenarios, therefore impacting the potential benefits that can be obtained by residents. This paper investigates these factors by identifying high-resolution greening scenarios that are able to maximize the cooling benefits for people and local residents. For the case study of metropolitan areas of Catania (Italy) with a hot Mediterranean climate, scenarios are derived by modelling physical and socio-economic factors as spatial constraints with the UMEP model and GIS spatial analysis. Results show that new greenery should be mostly located in public areas that are mostly used by residents. Built on the results obtained in the case study analyzed, the paper also proposes some general planning criteria for the localization of new urban greenery, which should be extended to other geographical urban contexts.

**Keywords:** climate simulations; climate regulation; urban planning; UMEP

## 1. Introduction

Urban systems are complex thermodynamic systems that are far from thermodynamic equilibrium and import energy, matter, and information from outer sources and dissipate heat as the result of different activities that make use of energy [1]. Urban environments are increasingly characterized by different issues related to climate change with global and local patterns, such increasing temperatures and pollution degrees and an increased quantity of water run-off and decreased quality of stormwater [2,3]. Such issues pose dramatic environmental and public health issues, impacting cities with increased frequency in the last decades [4,5], especially for highly vulnerable social subjects [6].

The positive role of urban vegetation in addressing the abovementioned urban issues is well known, as demonstrated by the rich and still-growing body of research, but also implemented urban projects and policies [7,8]. When strategically planned and designed in an integrated green infrastructure, urban vegetation and related ecosystems have the capacity to deliver a full array of ecosystem services, with direct and measurable benefits to urban communities and their well-being [9–11]. In the last years, the awareness of residents and city users of the importance of the services provided by urban vegetation has sharply increased so that requests for projects and tangible actions toward greener cities and more ecological neighbors are intensifying [12].

Among the many ecosystem services, climate regulation is of utmost importance in cities and represents the high concern of residents, who are increasingly asking for public

policy and actions to improve the local climate conditions of urban environments [13]. In Mediterranean geographical context, the rate of mortality due to heat waves has shown an increasing trend in the last 10 years. For example, data for 2021 in regions of south Italy revealed an increase of 15% in mortality [14].

Climate regulation of urban ecosystems is achieved by three main functions: physical shading of elements of the built environment; the modification of the flow of air; and the decrease in outdoor air temperature by evapotranspiration processes [15]. These functions generate relevant positive impacts on the energy consumptions of buildings that are directly shaded by trees and cool the air around buildings, with the effect of reducing the need of energy for cooling inside the buildings [16,17].

The positive effects of urban vegetation can be even more significant for elements of the urban environment that are directly and daily used by people. These include, streets, sidewalks, parking areas, and all different types of open spaces. Correct planning and design strategies of new greening can therefore reduce pedestrians' and other city dwellers' heat exposure thanks to the shading, transpiration, and wind-breaking functions [18,19] while also contributing to keeping impervious surfaces cooler, so they can emit less longwave radiation [20].

For these reasons, planting of urban trees is becoming a crucial planning and design strategy to reduce the excess heat in contemporary cities [21], and choosing the most effective spatial configurations of street trees may help optimize reductions of excessive heat, for example, by focusing on the street locations most in need of tree shade [22].

Climate simulations are modelling tools to the evaluate environmental behavior of specific environments (buildings and urban environments) and therefore inform planning choices on policy for urban greening and the localization of new vegetation, for example, by identifying portions of cities most in need of tree shade. Many researchers have explored the optimization of the location of trees shapes to reduce urban heat and mean radiant temperature according to specific urban geometries [23,24], and very recently, studies have specifically targeted the positive effects of new trees on pedestrians [20,25]. Furthermore, most of the research focuses on the spatial optimization of urban greenery, mainly at a very local scale (i.e., single building) [20] or at a wider regional scale [26].

However, limited efforts have been made to integrate results from these advanced pieces of research with more practical indication for urban planning [27] while taking into account the real opportunities offered by urban environments and morphologies [28]. More specifically, the integration of physical and socio-economic factors acting as important constraints in the planning and design of new urban vegetation remains unexplored. Such factors include land tenure, actual conditions of land use and land cover, and the possibility of generating benefits for a large number of residents. This paper thus proposes a spatially explicit method to identify high-resolution scenarios that are able to maximize the cooling benefits of urban vegetation for people and local residents by modelling physical and socio-economic factors as spatial constraints for the localization of new vegetation. At the same time, this research proposes planning criteria for the localization of new urban greenery that are built on the results obtained in the case study analyzed.

Section 2 presents the case study and data/material used. Section 3 introduces the methodology used, based on climate simulation and socio-environmental spatial analysis on the characteristics of the urban environment. Section 4 presents the results obtained with related maps, while Section 5 discusses them in the light of other relevant literature and recent research. Finally, the objectives of the research, main findings, and possible future work are summarized in Section 6.

## 2. Case Study and Material

The method was applied in three peri-urban areas located in the metropolitan area of Catania, Sicily, Italy (Figure 1). The metropolitan areas is the largest conurbation in the region, accounting for more than 700,000 residents in 27 municipalities. More than 60% of the total residents live in the municipalities surrounding the main city of Catania, which

has seen a progressive move of the population from the main city to smaller municipalities of the metropolitan area.

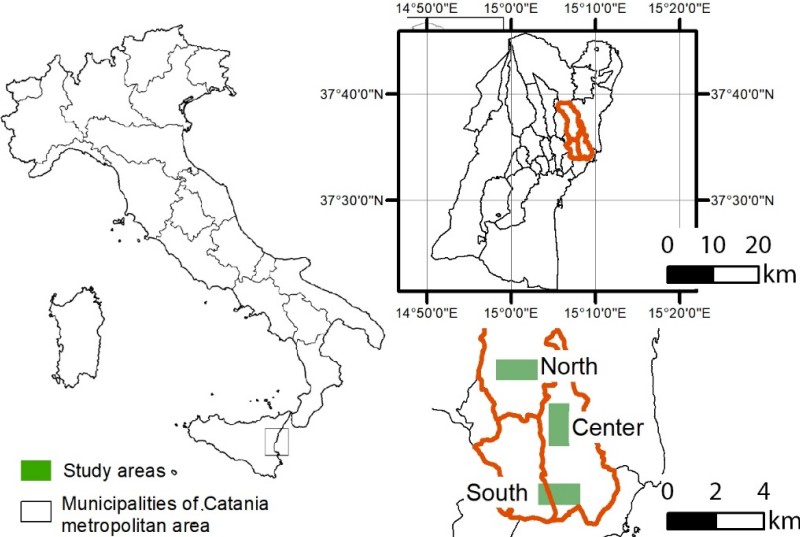

**Figure 1.** Location of the three case studies (north, center, and south) in the metropolitan area of Catania in Sicily (Italy).

These areas were chosen as they include different types of land use and land covers (Figure 2) with differentiated residential complexes, ranging from isolated villas to semi-detached houses and multi-story apartment complexes, which are typical features of European metropolitan areas [29]. Such variety of land use and land cover is functional in the evaluation of the impact of planned greening scenarios on different types of urban environments and is also useful for identifying different design options.

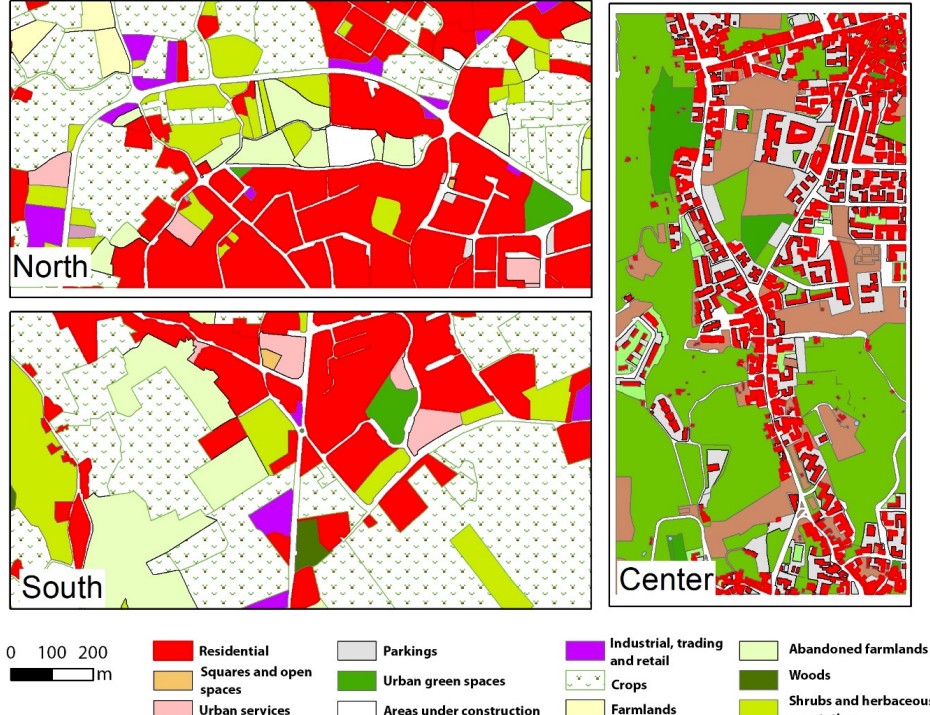

**Figure 2.** Land use of the three case studies.

From a climatic point of view, the Catania metropolitan area has a hot-mild Mediterranean climate (37.62° N, 15.17° E, 50 m a.s.l., annual average temperature of 17 °C) with hot summer temperatures very often above 30 °C, combined with a strong solar radiation of about 800 W/m$^2$ [16]. The close presence of the Mediterranean sea tends to ease natural ventilation, produces local cooling effects. In such climate conditions, the shadow effect of vegetation is a crucial factor that should be increased as much as possible by appropriate greening interventions in the built environment [16].

## 3. Method—Planning Criteria for Greening Scenarios

The methodology for the identification of greening scenarios is based on three main criteria that interact in the overall choice for the location of new greenery, with the aims of generating higher benefits for residents and, at the same time, ensuring a physical and socio-economic viability. The criteria are as follows: identification of areas with high outdoor thermal stress, physical/social feasibility of greening scenarios, and the maximization of the number of potential beneficiaries. The integration of these criteria identifies suitable areas for the planning of new greening scenarios. The criteria are illustrated in the next sub-sections.

### 3.1. High-Resolution Climate Simulations

The first step of the method identifies the areas with the most unfavorable conditions in terms of outdoor thermal comfort. They were selected by the evaluation of outdoor comfort in a reference condition of an hot summer day by the use of the Urban Multi-scale Environmental Predictor (UMEP) model.

UMEP is an integrated, open-access set of tools and models for urban climatology and climate-sensitive planning application, whose applications are mainly related to outdoor thermal comfort, consumption of urban energy, and climate change mitigation [30]. The most important feature of the model is its complete integration in the Geographical Information System (specifically QGIS open-access software). This allows users to use in a spatially explicit way all parameters of the model and, more importantly, to edit and map inputs and results directly in the GIS.

Among the different tools and models of UMEP, SOLWEIG (Solar and Long Wave Environmental Irradiance Geometry) is a model that simulates spatial variations of 3D radiation fluxes and Mean Radiant Temperature (Tmrt) in urban contexts [31]. Tmrt is one of the key meteorological variables accounting for energy balance and the thermal comfort of human beings, integrating shortwave and longwave radiation fluxes (both direct and reflected) to which the human body is exposed in outdoor and indoor environments [32]. In urban environments, Tmrt depends on building 3D geometries, street network, the albedo of building's facades, and land cover, and for this close relationship with urban morphology, it was chosen as a proxy of thermal comfort in this work. In SOLWEIG, Tmrt is derived by modelling shortwave and longwave fluxes in six directions (upward, downward, and from the four cardinal points) and angular factors.

To successfully perform a simulation, a set of information is requested by the model: meteorological data (global shortwave radiation, air temperature, and relative humidity), a digital surface model of buildings (including their height from ground) and vegetation, land cover above and below the canopy and maps of sky view factors. Vegetation and land cover (below-canopy land cover) maps are used to model the relationship between the built environment and vegetation so as to increase the accuracy of the results obtained. The reference day used for the meteorological data is the 9 July 1985, which was characterized by a maximum air temperature of 38°, humidity of 35%, and solar irradiance of 900 W m$^2$. The final outputs of SOLWEIG are raster maps of values of Tmrt for the reference day considered. The raster data necessary for the model are shown in Figure 3.

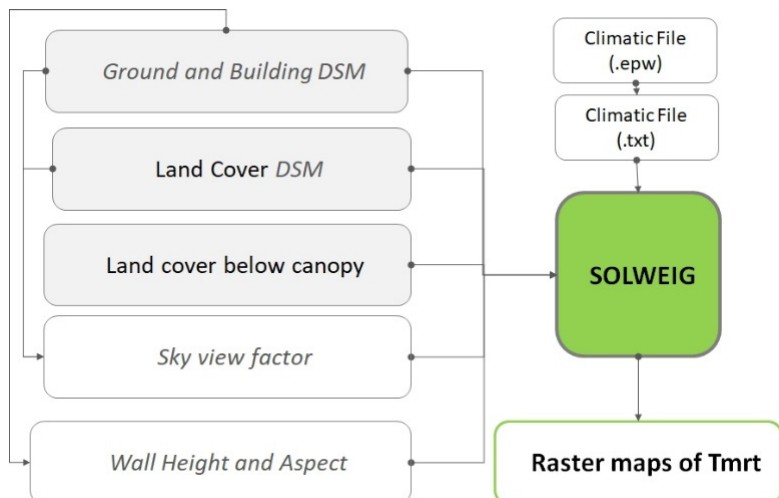

**Figure 3.** Input and output spatial layers in the SOLWEIG module of the UMEP model.

The above- and below-canopy land-cover values were analyzed by visual interpretation of high-resolution orthophotos (25 cm) available from regional cartography and further validated by comparison with photos from Google™ Street View. The following land-cover categories were identified: deciduous trees, evergreen trees, grass, paved areas, building, bare soil, and water. These categories are the land-cover types that are used as input layer in the SOLWEIG and, at the same time, represent typical land-cover classes that can be found in many urban and peri-urban contexts.

### 3.2. Physical/Social Feasibility of Greening Scenarios

The areas selected in the previous step were further evaluated in terms of social and physical constraints related to their potential transformation into green areas. The considered factors are the current land cover and land tenure. The first represents a constraint because greening scenarios can be developed on specific land-cover categories such as grass or bare soil, while other categories are unsuitable for the presence of buildings or trees (evergreen and deciduous trees) or less suitable due to their higher costs related to the planting of trees in paved and impervious areas.

Land tenure represents a socio-economical constraint because greening retrofitting in privately owned areas is more difficult to plan by public administrations and is often dependent on the willingness of single owners, although they can be subject to specific financial mechanisms to promote private greening interventions [33].

The categories of below-canopy land cover were further reclassified according to their suitability of being used as new areas to plant new trees in the greening scenarios. Suitable land-cover types included bare soil and grass but also paved areas belonging to roads' sideways or public squares. Despite the higher costs of implementation, some paved areas can be considered suitable for planting new trees because they are likely to generate benefits in terms of shading to higher number of people (see Section 3.3) [16].

Land tenure was derived by reclassification of available land-use maps in two main categories, namely private (mainly for residential/commercial and agricultural land-use categories) and public (for roads, public areas, parks, and public facilities). The land-use layer was derived from the urban atlas [34] of the metropolitan area of Catania, which was validated by visual interpretation of the available aerial photos.

The two vector layers derived by the re-classifications of land cover and land use are intersected in the next step of the method (Section 3.4) with the raster maps obtained in the climate simulations described in Section 3.2 to identify the proposed greening scenarios.

### 3.3. Maximization of the Number of Beneficiaries from Greening Scenarios

The third criterion considered was the potential usability of the built environment by pedestrians moving in the city, as they are the social subjects that can directly benefit from the presence of trees, especially when they make use of important urban elements such as sideways or other public spaces [35]. We analyzed the most highly used public areas so as to identify the areas where new greening scenarios could maximize their cooling effects on a higher number of beneficiaries. Such areas include highly used roads (including space for sideways, where pedestrian movement can occur), public spaces, and parking lots.

The most-used roads are considered to be those with the highest level of traffic, as traffic conditions are often related to the central localization of roads and high density of urban uses [36,37]. The roads with the highest level of traffic were selected by using traffic data extracted by the World Traffic Service of Arcgis™ Online Resources on a typical working day (13 October 2022). This service includes different types of traffic data, such as historical, live, and predictive traffic [38].

Other public spaces with a high level of usability (squares, green spaces and other public open spaces, and parking areas) were selected from the available land-use and land-cover vector layers.

### 3.4. Identification of the New Greening Scenarios

Greening scenarios are intended as spatial configurations of new trees to be planted in the three sub-areas and were obtained by the spatial integration of the three planning criteria introduced in the previous sections. Two scenarios were designed.

The first greening scenario involves public areas only. From the climate simulation performed with the SOLWEIG module of UMEP model, the areas with Tmrt values above 70° were selected, as these values represent a typical threshold value for outdoor thermal discomfort, and therefore, these areas represent prior targets for greening actions. From the criterion of physical/social feasibility of greening scenarios, pervious land (belonging to land-cover categories of bare soil and grass) and public areas were selected. From the criterion of the maximization of the number of beneficiaries, streets with a high level of traffic and all public areas were selected.

The second scenario was designed by adding private areas with pervious land covers to the areas already selected in the first scenario.

The vector layers expressing the above conditions for the three criteria were spatially overlaid, and the result was further refined by visual analysis and deleting or adjusting specific unsuitable situations (i.e., street with no space for sidewalks or areas with no physical accessibility by pedestrians). Table 1 summarizes the specific conditions of the three planning criteria that were used for the design of the greening scenarios.

**Table 1.** Criteria for the identification of the greening scenarios.

| Criterion | Scenario 1 (Public Areas) | Scenario 2 (Public + Private Areas) |
|---|---|---|
| *Identification of areas with high outdoor thermal stress* | Areas with median radiant temperature > 70° | Areas with median radiant temperature > 70° |
| *Physical/social feasibility of greening scenarios* | Land-cover categories<br>- Bare soil<br>- Grass<br>Land tenure:<br>- Public | Land-cover categories<br>- Bare soil<br>- Grass<br>Land tenure:<br>- Public<br>- Private |
| *Maximization of the number of beneficiaries of the greening scenario* | Highly used street<br>Public areas:<br>- Squares<br>- Open spaces<br>- Parking areas | Highly used street<br>Public areas:<br>- Squares<br>- Open spaces<br>- Parking areas |

Finally, new simulations on the two newly designed greening scenarios were performed with SOLWEIG module of the UMEP model to quantitively evaluate the resulting changes in Tmrt. Specifically, new evergreen trees with an average height of 9 m and canopy volume equal to 25% of the total volume of each tree (default option) were simulated in SOLWEIG. These trees are located in the areas identified in the two scenarios.

## 4. Results

### 4.1. Areas with High Outdoor Thermal Stress

Figure 4 maps the above-canopy land cover of the three areas. As already discussed, land-cover information is one of the input layers of the SOLWEIG module of the UMEP model.

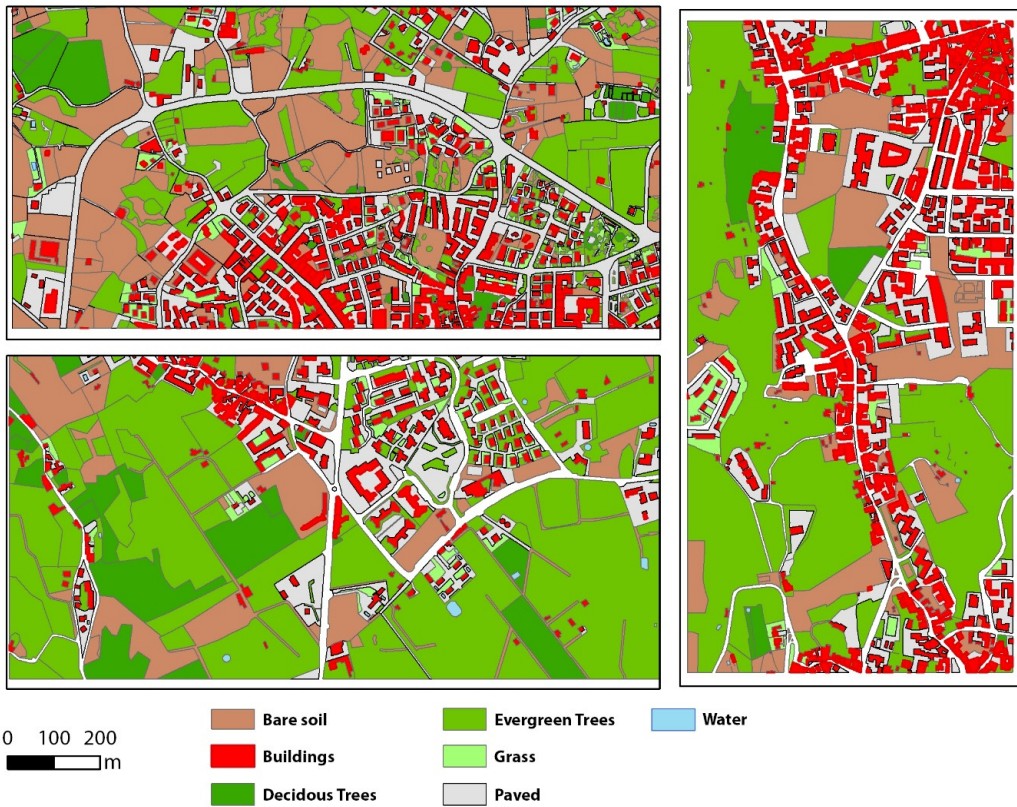

**Figure 4.** Maps of land cover for the three study areas.

Results from the simulation of the current situation are shown in Figure 5. Because of the considered very hot day on a summer afternoon, many pixels with very high values of Tmrt (above 80°) can be found, with similar values in the three areas. The share of values of Tmrt (classified by equal interval) is reported in Table 2 and Figure 6.

**Table 2.** Share of TMR values in the three areas.

| Area 1 (South) | | Area 2 (Central) | | Area 3 (North) | |
|---|---|---|---|---|---|
| Min: 40; Max: 85 Mean: 55 Std. Dev: 14 | | Min: 39; Max: 87 Mean: 63 Std. Dev: 14 | | Min: 38; Max: 86 Mean: 63 Std. Dev: 13 | |
| 40–50 | 52% | 39–50 | 35% | 38–50 | 27% |
| 50–60 | 3% | 50–60 | 11% | 50–60 | 6% |
| 60–70 | 10% | 60–70 | 1% | 60–70 | 16% |
| 70–85 | 33% | 70–87 | 53% | 70–86 | 51% |

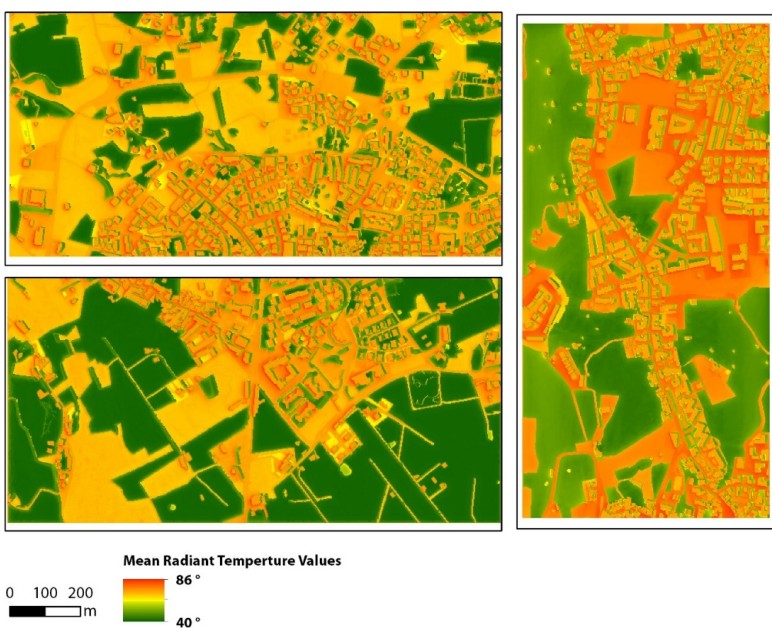

**Figure 5.** Results from the climate simulation for the current scenario.

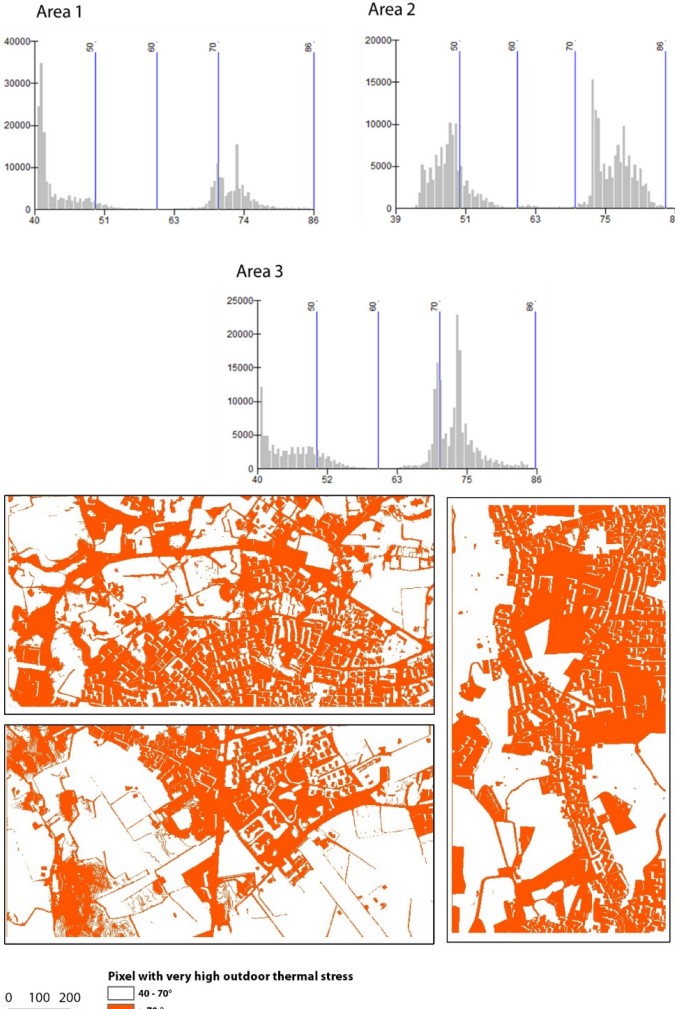

**Figure 6.** Distribution of Tmrt for the three areas in the current scenario. Pixels with Tmrt higher than 70°.

Pixels with Tmrt higher than 70° are present in relevant portions of the areas and are mapped in Figure 6. They represent 33% for area 1, 53% for area 2, and 51% for area 3, with an average of 45%.

All the three areas show similar distribution of values and are concentrated on lower values (around 40–45°) characterizing rural context and extreme values (around 70–75°) for buildings, streets, and paved areas but also bare soil. Being the most rural, area 1 has average and maximum values of TMR lower than the other two areas.

### 4.2. Physical/Social Feasibility of Greening Scenarios

Figure 7 maps the physical constraints related to the suitability of land-cover categories that can be used as new areas to plant new trees in the greening scenarios. Suitable land-cover categories sum up to 61% of area 1, 55% of area 2, and 54% of area 3. Figure 8 maps the land tenure for the three areas, with public areas accounting for 15% in area 1, 23% in area 2, and 11% for area 3.

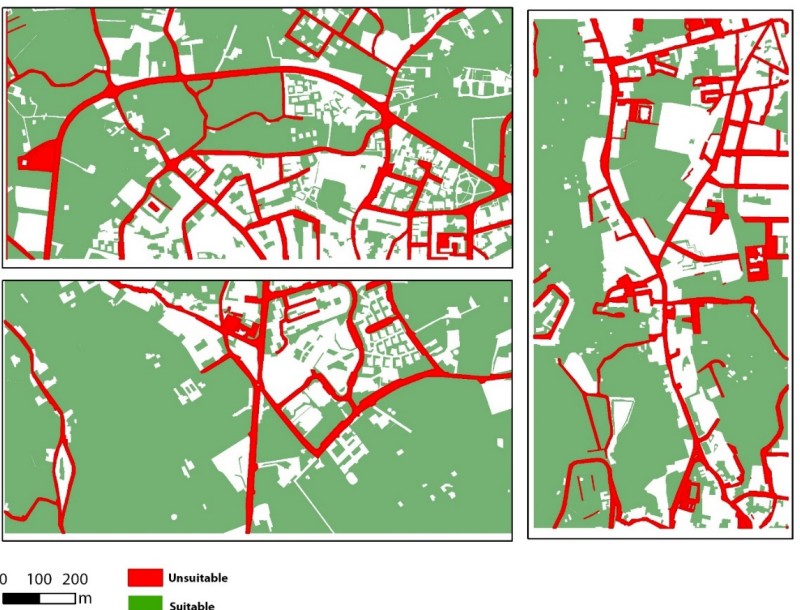

**Figure 7.** Physical constraints for greening scenarios: suitable land-cover categories.

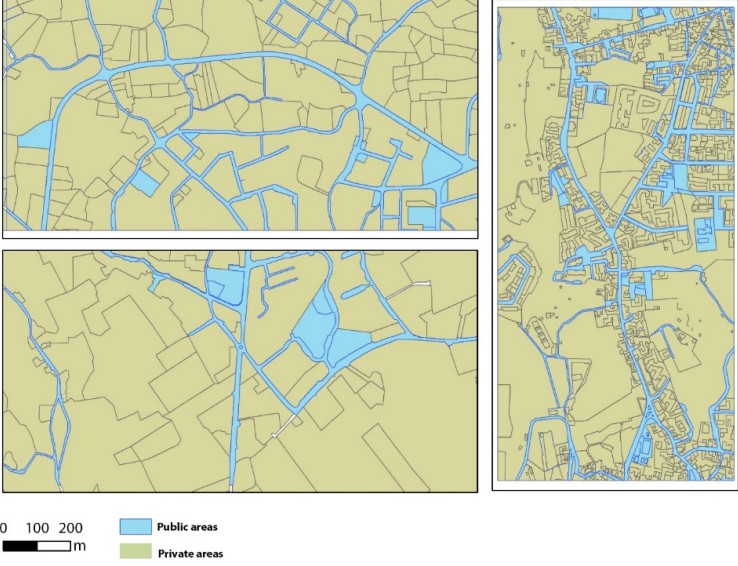

**Figure 8.** Social constraints for greening scenarios: land tenure.

### 4.3. Areas That Maximize the Number of Beneficiaries

Figure 9 maps the most-used spaces of the three areas (highly used streets with space for sideways and other public areas) where the new greenery can generate potential benefits to a high number of pedestrians and users of the urban environment.

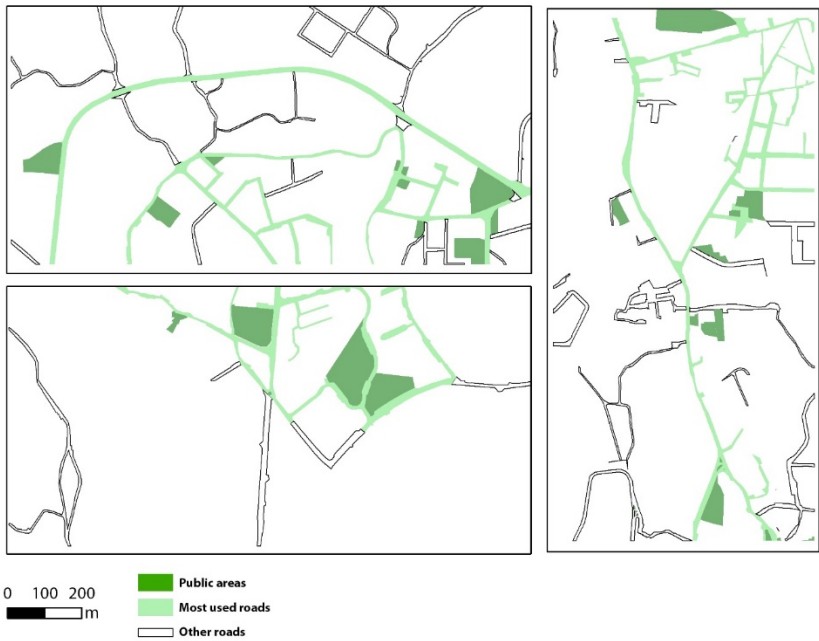

**Figure 9.** Highly used streets, squares, green spaces, and other public areas.

### 4.4. Greening Scenarios and New Simulations

The planning criteria reported in Table 1 were applied to the geographical layers presented in the previous Section 4.1, Section 4.2, and Section 4.3, following GIS geoprocessing and further local refinement of the results obtained by visual validation using Google Street View.

Figures 10 and 11 map the two scenarios obtained after Table 1, where the areas highlighted in cyan are the portion of the urban environment where new trees can be located. Scenario 1 includes public areas only, while scenario 2 includes both public and private areas. The extent of new greenery in the three areas is reported in Table 3. Area 1 is the one where the higher amount of new greening can be planned for the two scenarios. This is mainly due to the presence of an urban park, which is central to the area, that can be retrofitted with a good amount of new trees. Areas 2 and 3 show similar results, and new greening is mainly located along the sideways of the most-used streets and, to a more limited extent, in other public or private areas (squares, small urban parks, and private courtyards).

**Table 3.** Amount of new greenery in the 3 areas.

| | Area 1 (South) | | Area 2 (Central) | | Area 3 (North) | |
|---|---|---|---|---|---|---|
| | *Extent of New Greening ($m^2$)* | *% Total Area* | *Extent of New Greening ($m^2$)* | *% Total Area* | *Extent of New Greening ($m^2$)* | *% Total Area* |
| Scenario 1 (Public areas) | 22,646 | 9, 2 | 6992 | 2, 9 | 5534 | 2, 3 |
| Scenario 2 (Private + Public areas) | 39,627 | 16, 2 | 9906 | 4 | 13,109 | 5, 4 |

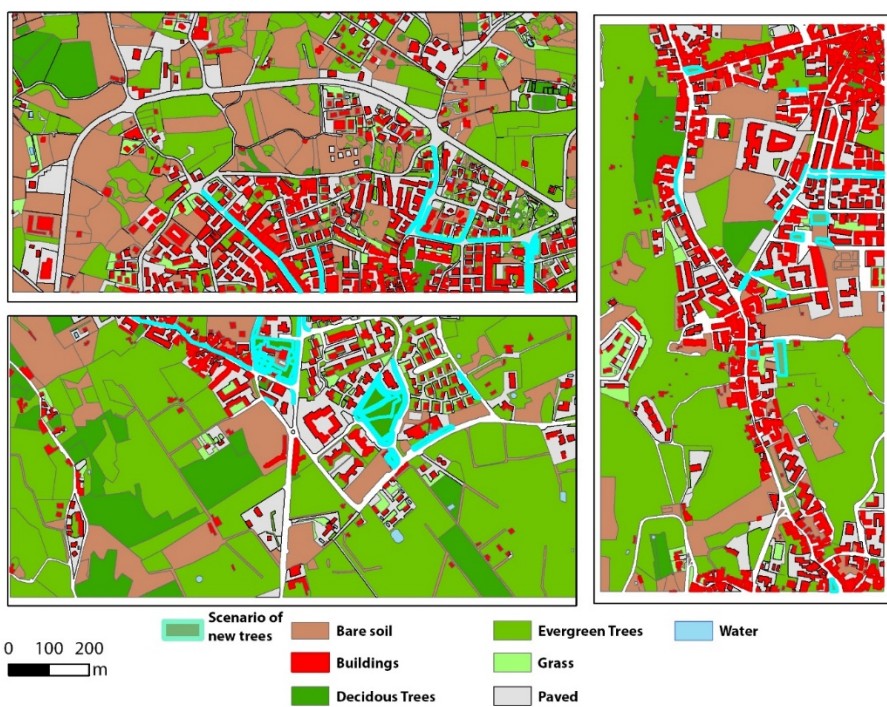

**Figure 10.** New greening scenarios in public areas (scenario 1).

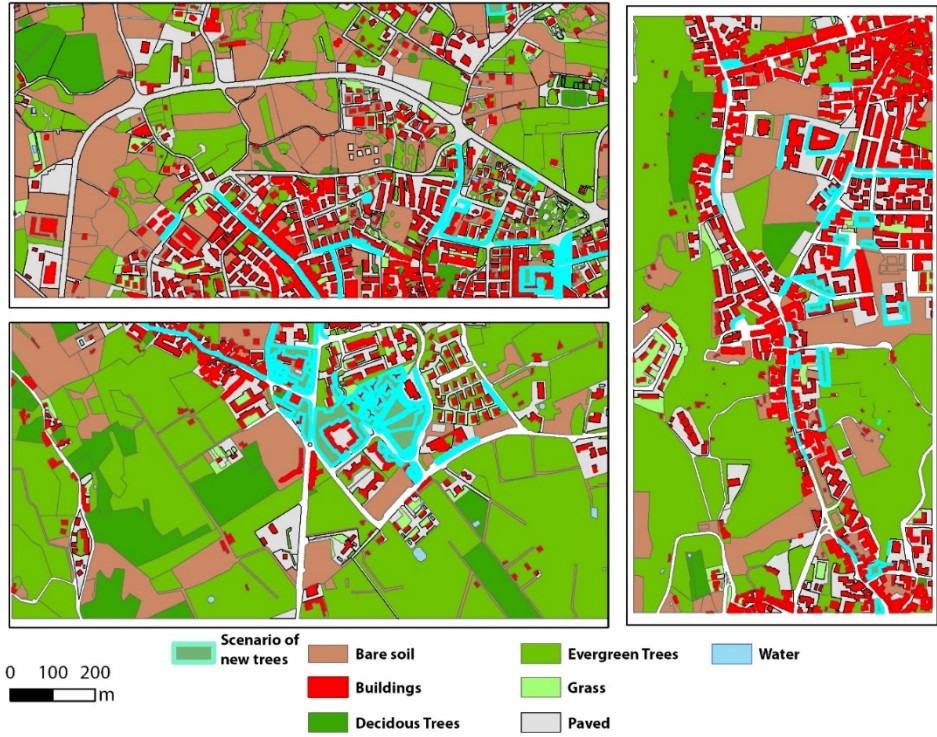

**Figure 11.** New greening scenarios in private and public areas (scenario 2).

The maps of Tmrt simulated from the two greening scenarios are shown in Figures 12 and 13 for scenarios 1 and 2, respectively.

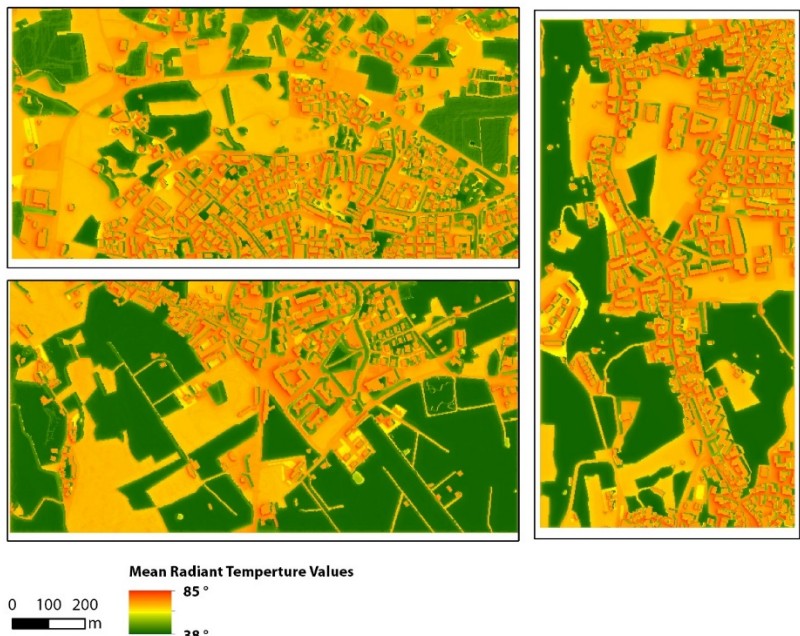

**Figure 12.** Results from the climate simulation for the scenario 1—public areas.

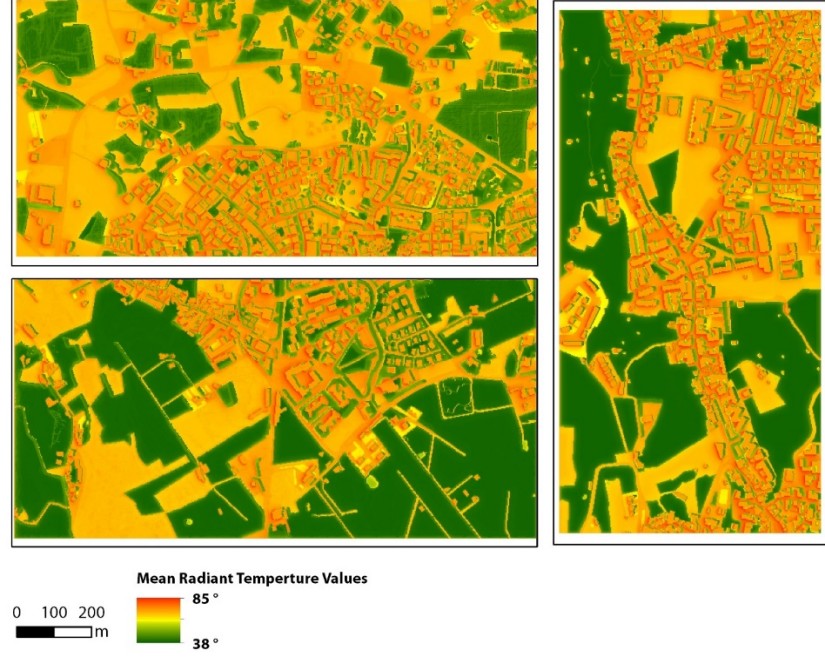

**Figure 13.** Results from the climate simulation for the scenario 2—public and private areas.

Changes in values of Tmrt with respect to the current situation are almost entirely localized in the space occupied by the new trees or in the areas that benefit from the direct shade of the trees. A very limited effect of the mass canopy can be seen from the resulting reduction of Tmrt. Table 4 reports the share of areas belonging to four classes of Tmrt values. The positive contribution of the new greening scenarios is visible yet limited: the comparison with the current scenario (Table 2) highlights that the amount of area with critically high Tmrt (>70) slightly decreases, and the area belonging to classes with the lowest Tmrt (40–50 and 50–60) increases by 2–3%.

**Table 4.** Tmrt values in the three areas for the two scenarios simulated.

| Area 1 (South) | | | Area 2 (Central) | | | Area 3 (North) | | |
|---|---|---|---|---|---|---|---|---|
| Min: 40; Max: 84 | | | Min: 39; Max: 87 | | | Min: 38; Max: 86 | | |
| Mean: 55 | | | Mean: 63 | | | Mean: 63 | | |
| Std. Dev: 14 | | | Std. Dev: 14 | | | Std. Dev: 13 | | |
| *Tmrt Values* | *Scenario 1* | *Scenario 2* | *Tmrt Values* | *Scenario 1* | *Scenario 2* | *Tmrt Values* | *Scenario 1* | *Scenario 2* |
| 40–50 | 54% | 55% | 39–50 | 36% | 39% | 38–50 | 28% | 29% |
| 50–60 | 4% | 3% | 50–60 | 6% | 6% | 50–60 | 8% | 6% |
| 60–70 | 10% | 11% | 60–70 | 12% | 10% | 60–70 | 17% | 20% |
| 70–85 | 32% | 30% | 70–87 | 45% | 45% | 70–86 | 47% | 45% |

To highlight the spatial differences of Tmrt values between the current and the greening scenarios, Figure 14 shows a smaller portions of area 1 (south) where the new greening is concentrated. An example of a possible design for area 3 (north) related to scenario 2 is shown in Figure 15.

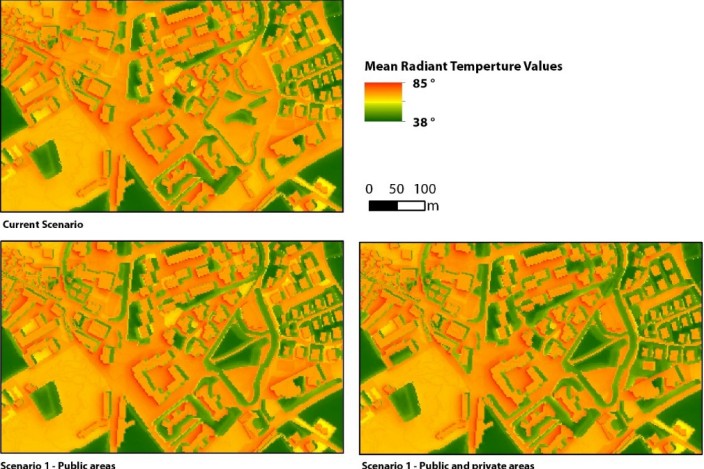

**Figure 14.** Excerpt from the simulations of Tmrt values for the greening scenarios in area 1 (south).

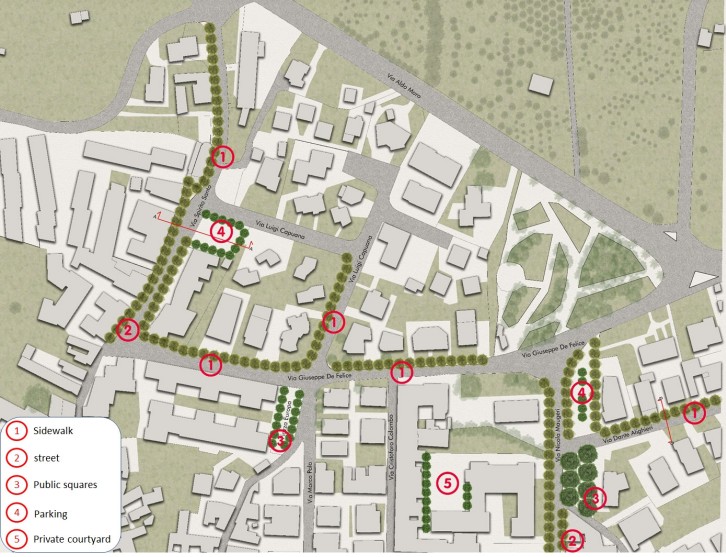

**Figure 15.** Design of greening scenario 2 (public + private areas) for area 2 (north).

## 5. Discussion

### 5.1. Evaluating the Effectiveness of Greening Scenarios in Decreasing the MRT

The main reason for the limited reduction in the average values of Tmrt is due to the location of the greening scenarios, mainly concentrated in the most urbanized areas, where the highest values of Tmrt can be found due to the presence of impermeable surfaces and buildings. Similar values have been reported in the same metropolitan area of Catania (Italy) by [39]. The reduction of Tmrt may also seem more limited than in other studies [40], but this is mainly because of the small spatial extent of the two greening scenarios. However, it must be highlighted that the scenarios were identified by taking into consideration the actual physical and socio-economic constraints present of the urban environment: such constraints limit the possible choice for the areas to be included in the greening scenarios but return planning options that are closer to reality. This relationship with the actual urban environment and socio-economic condition represents the main element of novelty of this research.

### 5.2. Proposal of General Planning Criteria for New Urban Greening Based on the Performed Simulations

The results from the simulations of the current and new scenarios of urban greening indicate some useful planning criteria that can be generalized to different urban contexts with similar climatic characteristics and greening objectives. These criteria are about the type, location, and extent of the greening and the choice of the species that can be used.

First, simulations highlight that trees represent the most effective type of green elements in terms of cooling capacities, while the contribution of other types of greenery (shrubs, grass, and herbaceous vegetation) is more limited because of the reduced height and canopy volume. This is in line with other studies [40,41] and suggests that shrubs and grass can be used for ornamental or ecological purposes, i.e., for completing vegetation levels in parks or public spaces or to generate other important functions such as noise reduction or carbon sequestration [42].

Prior choices for the location of new trees should select public areas (sidewalks, streets, and public areas) with permeable land cover for the easier suitability and economic viability of tree planting. Areas at lower priority can include private areas, where landowners must be convinced or supported economically with specific incentives for planting trees [13]. Moreover, impervious land cover represents areas with lower suitability due to the higher costs for planting new trees. This highlights an important trade-off for current impervious street sideways between the maximization of the benefits that can be obtaining and the costs for planting new trees. Furthermore, areas close to buildings with aspects to the south and/or west can be chosen to maximize the shading potential of new trees [43,44].

The extent of new greenery is a variable depending on the cooling objectives assumed by the greening scenarios as well as the available economic resources. The extent of new greenery should be large enough to cover all areas of the urban environment under thermic stress (i.e., with Tmrt higher than an assumed threshold). The extent of the greenery should also include highly used portions of the city, such as streets and public areas, so as to maximize thermal comfort for people.

Given a specific climate and overall water availability, the most suitable species should have some characteristics in order to maximize their climate regulation potential: high-density canopy, deciduous species for hot climates, evergreen for cold climates, and fast-growing, cheap to plant and manage, and able to maximize the ecological heterogeneity of the urban context. However, in some cases, not-autochthonous species can be accepted as suitable species, as demonstrated by the widespread presence of Australian species in southern Italian urban contexts [45]. Possible suitable species for Mediterranean climates include, among others, *Pinus pinaster, Ficus benjamina,* and *Quercus ilex* [16,45].

### 5.3. Economical Viability

It is widely acknowledged that investments in urban greening can ensure positive returns on the quality of urban environments and human health and in economic terms [46–48]. However, specific political and economic factors might hamper the actual implementation of greening projects. The most common of these factors is the need for public acquisition of the land where the greening scenarios should implemented because, often, public administrations do not own such land. The availability of economic resources required to buy land from private landowners thus represents a strong constraint for the final implementation and maintenance of greening projects. To face this lack of economic resources, public administration must find alternative sources of funding or design incentive-based mechanisms, compensations for land acquisition, and payment for ecosystem services [49,50].

For this reason, public areas (i.e., streets, sidewalks, and open spaces) represent high-priority areas in which to concentrate economic resources for the implementation of greening scenarios because they do not need to be acquired.

As already stated, the greening scenarios have been identified with the main objectives of improving the outdoor thermal comfort of the urban environment, but it must be underlined that these scenarios are able to generate many other positive contributions to an urban environment and its residents. For example, in terms of regulation of urban water run-off and pluvial flooding reduction, new trees located along streets would be crucial elements [10,51].

### 5.4. Limitations

There are some limitations that can be acknowledged in this research. First, the scenarios of reduced values of Tmrt are missing a ground validation, which can be very useful for better tuning the model results to the actual conditions of the urban environment under analysis and therefore identifying more reliable and effective greening scenarios. However, this process can be long and complex, as it would require identifying existing green elements in specific parts of cities (i.e., street trees or trees on sidewalks) to be used as a reference configuration for the ground measurement of Tmrt with radiometers. Results from these measurements of Tmrt can then be compared with the values of simulated Tmrt for scenarios with similar configuration of green elements. With this approach, it would be possible to better understand the difference between the simulated and measured values of Tmrt.

The SOLWEIG module has specific characteristics and simplifications. In its formulation, it does not evaluate the contribution of evapotranspiration of vegetation, which is a relevant benefit that can be directly perceived by people who are walking or moving close to trees. It makes use a single value for the albedo for all the facades of buildings, and it also does not allow the modelling of the contribution of shrubs but only of trees and grass, so other forms of urban greenery such as green roofs and walls, which are important options for urban planners and designers, cannot be included in simulations.

Finally, the results of the simulations used to identify urban areas with high levels of thermal stress referred to the hottest hour of the day (3 p.m.) (see Figure 6). In other times, different areas can show high values of thermal stress so that different greening scenarios can be identified. However, from our analysis, the total amount of areas with high levels of thermal stress were at their maximum at 3 p.m., so this hour can be considered as a reliable reference condition because it represents the worst thermal situation to be addressed with greening scenarios.

## 6. Conclusions

This paper proposes a spatially explicit method to identify planning scenarios to maximize the cooling benefits of urban vegetation for people and local residents, which represent the main beneficiaries of any public policy and action for climate regulation. Built

on high-resolution simulations of mean radiant temperature and modelling of physical and socio-economic factors acting as spatial constraints, two greening scenarios were designed.

The case studies are three peri-urban areas located in the Catania metropolitan areas in Sicily, characterized by a hot and dry Mediterranean climate and exposed to several extreme hot days during the summer.

The proposed greening scenarios represent feasible and viable planning options, where the positive contribution of trees in reducing the values of Tmrt is concentrated to the areas where the trees can be planted. This outcome highlights the importance of developing greening projects targeting the highly used areas of the built environment, such as sideways and other public spaces. This is very relevant, especially for public administrations with limited economic resources to implement these scenarios.

The integration of the actual physical and socio-economic conditions of the urban environment and the high resolution of the analysis in identifying the greening scenarios represent the main elements of novelty of this research.

Based on the results obtained, the general planning and design criteria were proposed with regards to the type, location, and extent of the new greening and the choice of the tree species. Such criteria can be used to maximize the benefits of climate regulation while at the same time ensuring the actual socio-economic viability of the identified options of urban greening.

Further research can be oriented toward the integration of the contribution of evapotranspiration in the evaluation of outdoor comfort by using other spatially explicit models and economic valuations of the greening scenarios to dynamically understand the correct return of public investments over a designed period of time.

**Author Contributions:** Conceptualization, D.L.R.; Methodology, D.L.R. and J.L.; Software, D.L.R.; Validation, D.L.R.; Formal analysis, D.L.R.; Resources, D.L.R.; Data curation, D.L.R.; Writing—original draft, D.L.R. and J.L.; Writing—review & editing, D.L.R. and J.L.; Visualization, D.L.R.; Supervision, D.L.R. and J.L. All authors have read and agreed to the published version of the manuscript.

**Funding:** This research received no external funding.

**Institutional Review Board Statement:** Not available.

**Data Availability Statement:** Not available.

**Conflicts of Interest:** The authors declare no conflict of interest.

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
