# Peer review of "High-Resolution Greening Scenarios for Urban Climate Regulation Based on Physical and Socio-Economical Factors"

_sustainability, doi:10.3390/su15097678_

Round 1

Reviewer 1 Report

Dear authors.

I am grateful to review the article "High Resolution Greening Scenarios for Urban Climate Regulation Based on Physical and Socio-Economic Factors".

The article is interesting and relevant. In general, I rate the article positively. However, I have some questions:

1. Why is there 4 key areas shown in Figure 1, and only 3 in Figures 2 and the rest? Specify the numbers of the key sections.

2. Improve the quality of figure 1. Perhaps split it into 2 drawings. Sign geographical objects – countries, seas. Show the position of the object under study on the map of the world, Europe.

3. In Figures 1-9, specify the coordinates and the coordinate grid.

4. Make an article according to the rules of the journal.

5. From which sources your materials were obtained. What is the difference between Figures 2 and 3?

6. What does "Tmrt" mean? Why don't you use an abbreviation? Clarify this point. Is it more appropriate to write in full?

7. Describe the methodology in more detail so that other researchers can reproduce it step by step.

8. I recommend shortening section 6 a bit.

The article can be accepted after the elimination of comments.

Author Response

Detailed responses to reviewers’ comments to the previous submissions follow below. The number of pages and lines indicated in the responses refer to the new revised version of the manuscript, where the main changes have been highlighted in green.

Dear authors.

I am grateful to review the article "High Resolution Greening Scenarios for Urban Climate Regulation Based on Physical and Socio-Economic Factors".

The article is interesting and relevant. In general, I rate the article positively. However, I have some questions:

  1. Why is there 4 key areas shown in Figure 1, and only 3 in Figures 2 and the rest? Specify the numbers of the key sections.

Author’s comment:

We are sorry for the mistake, fig. 1 has been corrected, showing the 3 study areas, with the indication of north, center and south.

Figure 2 has also been updated with the specification of the 3 study areas.

  1. Improve the quality of figure 1. Perhaps split it into 2 drawings. Sign geographical objects – countries, seas. Show the position of the object under study on the map of the world, Europe.

Author’s comment:

Fig. 1 intends only to locate the case study in Italy, so we think that location of Italy within Europe would not really represent an addition information of meaningful value.

Furthermore, there are already many maps in the paper, we prefer to not add another one.

  1. In Figures 1-9, specify the coordinates and the coordinate grid.

Author’s comment:

 We added coordinate grid in fig. 1. For the high-resolution nature of this study, all the other maps show small areas where grid reticules would look a redundant and not adding relevant additional information.

  1. Make an article according to the rules of the journal.

Author’s comment:

 Article has been formatted according the journal’s requirement.

  1. From which sources your materials were obtained. What is the difference between Figures 2 and 3?

Author’s comment:

 The source for the materials and data used is reported in section 2.

Figure 2 maps the land use pattern of the study areas, to show the extent of urbanization and non-urbanised areas, while figure 3 is a kind graphical abstract of the functioning of the SOLWEIG module of UMEP model.

  1. What does "Tmrt" mean? Why don't you use an abbreviation? Clarify this point. Is it more appropriate to write in full?

Author’s comment:

 Tmrt is the Mean Radiant Temperature, as indicated for the fist time in page 5, lines 149-151.

The acronym is quite useful this time, as it is used many times along the text.

  1. Describe the methodology in more detail so that other researchers can reproduce it step by step.

Author’s comment:

 All steps have been

  1. I recommend shortening section 6 a bit.

Author’s comment:

Some paragraphs and lines has been reformulated in a shorter form, especially the ones in page 16, lines 357-360, lines 414-416, lines, 420-422, 445-448.

Author Response

Some of the comments for the authors are as follows:

  • Abstract, Introduction and Case study review are written in a proper way. It is well-explained why case studies of the municipalities surrounding the main city of Catania were taken into conisderation. Abstract, Introduction and Case study review include relevant sources as well as basic explanations of the posibilities for urban climate regulation
  • Methodology is adequate and reliable for this kind of research. The authors used combination of several critetia in order to illustrate posibilities for urban climate regulation and they provide reader with the arguments that are in accordance with the methodology and exepected results.
  • Results are interesting and they contain clear ideas of the problem. Regarding the presentation of results, the authors use several figures that show very clear guidelines of posible greening scenarios.
  • In spite of some limitations in the research, discussion is written in a proper way by explaining those limitations and by giving ideas for the future deliberations.
  • Conclusion is written correctly with clear ideas for the further researching steps.

Author’s comment:

 We would like to thank the reviewer for his/her appreciation of our work.

Reviewer 3 Report

The paper brings up a fantastic issue of urban greening scenarios taking the example of Italy. Authors can improve the paper by incorporating the following concerns.  

a.       Rearrange keywords alphabetically.

b.      Paragraph 6 of page 2 please rephrase the term “well developed researches”

c.       Merge paragraph 6 and 7 of page 2

d.      Authors should split sentences about organization of the study from Paragraph 7 page 2.

e.       What you mean by “the research question” in last sentence of your paragraph 7 page 2

f.        Author should precisely indicate/label the study areas on the figure 1. I suggest separating maps and use symbology on for showing what is where.

g.      What are the criteria/base to classify the land use/cover into different classes. At least you should use references for your classification of land use.

h.      I didn’t see your data source and description: of which year and why you use the data.

i.        What you mean by the “hot summer day” in your 3.1. section, when it was? I missed explanation of data acquisition.

j.        Too many figures in the paper. I suggest some figures to be at the appendix section.

k.      Your discussion about “economic feasibility” is unclear. 

l.        Correct your “fig” vs “figure” usage throughout the manuscript. Be consistent.

m.    What you mean by “ the spatial extent of areas with high levels of thermal stress resulted maximum at 3 pm, so this hour can be considered as a reliable reference condition.” Page 17

n.      The conclusion section in the paper has to better elaborate the main findings.

o.      List of references has to be corrected, e.g., reference 3 

Grammar 

Paragraph flow and structure  

Author Response

Reviewer 3

The paper brings up a fantastic issue of urban greening scenarios taking the example of Italy. Authors can improve the paper by incorporating the following concerns.  

  1. Rearrange keywords alphabetically.

Author’s comment:

 Edited accordingly

  1. Paragraph 6 of page 2 please rephrase the term “well developed researches”

Author’s comment:

 Term has been deleted for sake of clarity

  1. Merge paragraph 6 and 7 of page 2

Author’s comment:

 Edited accordingly (page 2, lines 87).

  1. Authors should split sentences about organization of the study from Paragraph 7 page 2.

Author’s comment:

 Edited accordingly (page 2-3, lines 95-99).

  1. What you mean by “the research question” in last sentence of your paragraph 7 page 2

Author’s comment:

 We meant the research objective of our work. We have rephrased it to make it clearer (page 3, lines 97)

  1. Author should precisely indicate/label the study areas on the figure 1. I suggest separating maps and use symbology on for showing what is where.

Author’s comment:

 Labels on the 3 study areas (north, central, south) have been added in fig 1  and fig. 2 to better identified them and figure caption has been updated.

  1. What are the criteria/base to classify the land use/cover into different classes. At least you should use references for your classification of land use.

Author’s comment:

The land use classes mainly follows the urban atlas categories, but with some validation. An explanation has been added in page 6, lines 202-204.

The choice of the land cover classes depended on the SOLWEIG module, as explained in page 5, lines 174-178

  1. I didn’t see your data source and description: of which year and why you use the data.

Author’s comment:

Data sources and description are reported in:

- Section 3.1 for for the aerial photo and UMEP inputs variables

- Section 3.2 for land use (see previous comment)

  1. What you mean by the “hot summer day” in your 3.1. section, when it was? I missed explanation of data acquisition.

Author’s comment:

That is the reference day that was used in the simulation. It is explained in the same section in page 5, lines 165-167

  1. Too many figures in the paper. I suggest some figures to be at the appendix section.

Author’s comment:

We agree with reviewers that there are many figures and maps, but we would prefer to provide an useful spatial evidences of all the results obtained. Since all maps follow strictly the methodology, it would not be possible to put some of them in appendix, as the sequential order of the results could be lost and not easy to be fully understood.

  1. Your discussion about “economic feasibility” is unclear. 

Author’s comment:

 This section has been renamed to “economic viability”, a better appropriate term. We here intended to discuss the main economic issues that could be raised when trying to physically implement one of the greening scenarios proposed.

We acknowledge that some sentence were not clear, so we have re-worded the related section (pag 17, lines 401-422).

  1. Correct your “fig” vs “figure” usage throughout the manuscript. Be consistent.

Author’s comment:

 Edited accordingly along the entire text.

  1. What you mean by “ the spatial extent of areas with high levels of thermal stress resulted maximum at 3 pm, so this hour can be considered as a reliable reference condition.” Page 17

Author’s comment:

 We have reworded the sentence (pag 17, lines 445-448).

  1. The conclusion section in the paper has to better elaborate the main findings.

Author’s comment:

More comments have been added in the conclusions for results and findings, and they are synthetized in page 18, lines 460-473.

  1. List of references has to be corrected, e.g., reference 3 

Author’s comment:

References has been updated according to journal standard.

Round 2

Reviewer 3 Report

I am satisfied with the changes made by authors in improving the manuscript. Authors recommended to:

Line 93 Separate the “Section 2 presents…” into a different paragraph.

Author Response

I am satisfied with the changes made by authors in improving the manuscript. Authors recommended to:

Line 93 Separate the “Section 2 presents…” into a different paragraph.

AUTHORS' Comment

Edited accordingly (now line 94).